# Update on the Management of Stage III NSCLC: Navigating a Complex and Heterogeneous Stage of Disease

**Arani Sathiyapalan** [1,2], **Ziad Baloush** [1,2] **and Peter M. Ellis** [1,2,*]

1   Juravinski Cancer Centre at Hamilton Health Sciences, Hamilton, ON L8V 5C2, Canada;
    sathiyapal@hhsc.ca (A.S.); baloush@hhsc.ca (Z.B.)
2   Department of Oncology, McMaster University, Hamilton, ON L8S 4L8, Canada
*   Correspondence: ellisp@hhsc.ca; Tel.: +1-905-387-9495

**Abstract:** Background: Stage III nonsmall cell lung cancer (NSCLC) represents a heterogeneous group of patients. Many patients are treated with curative intent multimodality therapy, either surgical resection plus systemic therapy or chemoradiation plus immunotherapy. However, many patients are not suitable for curative intent therapy and are treated with palliative systemic therapy or best supportive care. Methods: This paper is a review of recent advances in the management of patients with curative intent disease. Results: There have been significant advances in curative intent therapy for patients with stage III NSCLC in recent years. These include both adjuvant and neoadjuvant systemic therapies. For patients with resectable NSCLC, two trials have demonstrated that adjuvant atezolizumab or pembrolizumab, following chemotherapy, significantly improved disease-free survival (DFS). In patients with tumours harbouring a common mutation of the *EGFR* gene, adjuvant osimertinib therapy was associated with a large improvement in both DFS and overall survival (OS). Five randomized trials have evaluated chemotherapy plus nivolumab, pembrolizumab, durvalumab, or toripalimab, either as neoadjuvant or perioperative (neoadjuvant plus adjuvant) therapy. All five trials show significant improvements in the rate of pathologic complete response (pCR) and event-free survival (EFS). OS data are currently immature. This would now be considered the standard of care for resectable stage III NSCLC. The addition of durvalumab to chemoradiation has also become the standard of care in unresectable stage III NSCLC. One year of consolidation durvalumab following concurrent chemoradiation has demonstrated significant improvements in both progression-free and overall survival. Conclusions: Immune checkpoint inhibitor (ICI) therapy has become a standard recommendation in curative intent therapy for stage III NSCLC.

**Keywords:** NSCLC; neoadjuvant therapy; adjuvant therapy; chemotherapy; immunotherapy; radiation

## 1. Introduction

Lung cancer represents the most common cancer in Canada, of which about 85% are non-small cell lung cancer (NSCLC). Approximately 20% of NSCLC cases are stage III at the time of diagnosis [1]. These patients represent a very heterogeneous group, with multiple approaches to treatment that are highly dependent on both disease characteristics as well as patient prognostic factors such as Eastern Cooperative Oncology Group (ECOG) performance status (PS) and recent weight loss [2].

The majority of patients have stage III disease based on the presence of a large primary tumour, obvious hilar, or mediastinal nodal disease at the time of diagnosis (T3N1, T4N$_{any}$, T$_{any}$N2) [3]. A proportion of patients, though, are diagnosed with stage III NSCLC based on microscopic mediastinal lymph node disease identified following surgical resection. Some patients may have stage III disease that is considered surgically resectable, but historically, the majority of patients are considered unresectable and treated with radiation and/or systemic therapies. The definition of resectable stage III disease is not clearly

defined and very surgeon-dependent. However, it is generally limited to patients with adequate performance status, lower volume ipsilateral mediastinal lymph node disease, and adequate pulmonary function [4]. In this setting, multidisciplinary case conferencing may improve outcomes [5]. It is important to ensure all treatment options are considered and there is agreement from all disciplines regarding which patients with stage III NSCLC are considered surgically resectable.

Stage III NSCLC is generally treated with a multimodality approach. Many patients are treated with curative intent approaches, including surgical resection followed by adjuvant systemic therapy, or definitive chemotherapy plus radiation [6,7]. Neoadjuvant approaches, including chemotherapy or chemotherapy plus radiation, may be considered prior to surgical resection [8,9]. However, many patients are not suitable for curative intent therapy because of poor performance status, volume of disease, or comorbid health conditions and are treated with palliative intent systemic therapy or best supportive care. This paper represents an overview of current management approaches for curative intent stage III NSCLC. The focus is on recent advances for both resectable and unresectable diseases and some insight into future research directions.

## 2. Resectable NSCLC

Approximately 25–30% of patients with NSCLC are considered resectable and go on to receive curative intent surgery [10]. However, many of these patients develop recurrence due to the presence of micrometastatic disease, and survival rates range from 90% in stage IA to 36% in stage IIIA disease [3]. This section focuses on strategies to improve survival for those patients with resectable NSCLC. The majority of patients with stage I and II NSCLC are candidates for upfront resection unless medically unfit. However, controversy exists about the definition of resectable stage III NSCLC. Patients with stage III NSCLC without mediastinal nodal involvement (T3N1, T4N0-1) would generally be considered resectable so long as they are medically fit. Patients with pathologically involved mediastinal lymph nodes (TanyN2) are commonly deemed unresectable and discussed in the section below. A subset of these patients, though, with nonbulky (<2–3 cm) mediastinal lymph node involvement and a limited number of involved lymph node stations, may also be considered surgically resectable [11].

### 2.1. Adjuvant Systemic Therapy

Adjuvant systemic therapy with cisplatin doublet chemotherapy is routinely recommended following surgical resection of stage II–IIIB NSCLC and may be considered in patients with stage IB tumours $\geq$ 4 cm (AJCC 7th edition) [6,12,13]. Data from the lung adjuvant cisplatin evaluation (LACE) meta-analysis demonstrated a 5.4% improvement in five-year survival for patients receiving adjuvant platinum-based chemotherapy [14]. The magnitude of benefit in both disease-free (DFS) and overall survival (OS) was stage-dependent. There was a trend towards worse survival among patients with stage IA and IB (tumours < 4 cm) and increasing benefit with higher stage disease (11.6% and 14.7% improvement in five-year OS for stage II and III disease) [15,16]. The largest risk reduction was observed in trials evaluating cisplatin and vinorelbine; however, cisplatin and pemetrexed have demonstrated similar efficacy and lower toxicity in patients with nonsquamous NSCLC [17].

Adjuvant chemotherapy has represented the standard of care since 2005. In advanced NSCLC, the adoption of immune checkpoint inhibitors (ICI) has transformed the treatment and prognosis of the disease [18–21]. In recent years, new data from several randomized trials have demonstrated improvements in DFS from the addition of ICI to standard-of-care adjuvant systemic therapy in this population.

The IMpower010 trial, a phase III RCT, evaluated the addition of one year of adjuvant atezolizumab after completion of adjuvant platinum-based chemotherapy, as compared to standard of care adjuvant chemotherapy, in patients with resected stage IB (T ≥ 4 cm) to stage IIIA (AJCC 7th edition) NSCLC (Table 1) [22]. The trial included patients regardless of programmed death ligand 1 (PD-L1) expression; however, the primary outcome was disease-free survival (DFS) in the stage II-IIIA patients with PD-L1 expression > 1%. The risk of recurrence was reduced by 34% (HR 0.66, 95% CI 0.50–0.88), with an absolute improvement in DFS at three years of 12% (60% versus 48%). Improved DFS was observed in all patients with stage II-IIIA NSCLC (HR 0.79, 95% CI 0.64–0.96), as well as the intention to treat (ITT) population of stage IB-IIIA NSCLC (HR 0.81, 95% CI 0.67–0.99). The largest improvement in DFS was observed in a prespecified subgroup of stage II-IIIA NSCLC with tumour expression of PD-L1 ≥ 50% (HR 0.43, 95% CI 0.27–0.67). Post-hoc analysis suggested attenuated benefit in the PD-L1 1–49% group (HR 0.87, 95% CI 0.60–1.26) and minimal benefit in patients with tumour PD-L1 < 1% (HR 0.97, 95% CI 0.72–1.31). Overall survival was immature and noted a trend towards benefit in the ITT population; however, subgroup analysis revealed that this benefit was driven by the Stage IIIA PD-L1 ≥ 50% population. The toxicity profile was in keeping with the known safety profile of ICI agents, with an approximate 11% treatment-related Grade 3–4 adverse event rate. The discontinuation rate was slightly higher than ICI studies in the advanced setting (18%), suggesting a lower tolerance for adverse events in the curative-intent setting.

The US FDA approved adjuvant therapy with atezolizumab in patients with resected stage II-IIIA NSCLC and tumour expression PD-L1 ≥ 1%. In other countries, such as Canada, the approval of adjuvant atezolizumab was limited to patients with stage II-IIIA NSCLC and tumours with high PD-L1 expression ≥50%. There has been rapid adoption of these data into guidelines such as the joint American Society of Clinical Oncology (ASCO)/Ontario Health-Cancer Care Ontario (OH-CCO) guideline, which recommends adjuvant cisplatin-based chemotherapy followed by atezolizumab in patients with resected stage II-IIIA (7th edition) NSCLC and tumours expressing PD-L1 ≥ 1% [7].

Similarly, PEARLS/Keynote 091 was a phase III randomized trial evaluating one year of adjuvant pembrolizumab every three weeks in patients with resected stage IB (T ≥ 4 cm)—IIIA NSCLC (AJCC 7th edition) and any PD-L1 expression [23]. Adjuvant chemotherapy was not mandatory but was received by approximately 85% of patients. The primary endpoint of DFS in the overall population demonstrated a relative risk reduction of 24% (HR 0.76, 95% CI 0.63–0.91), which translated to an approximate 11-month improvement in median DFS (53.6 months vs. 42 months). Interestingly, PD-L1 was not a predictive biomarker in this setting, as benefit was seen in all PD-L1 subgroups (PD-L1 ≥50% HR 0.82, 95% CI 0.57–1.18, PD-L1 1–49% HR 0.67, 95% CI 0.48–0.92), PD-L1 < 1% HR 0.78, 95% CI 0.0.59–1.05). However, the subgroup of patients that did not receive adjuvant chemotherapy may not benefit from the addition of pembrolizumab (HR 1.25, 95% CI 0.76–2.05). This may be for a multitude of reasons, including patient stage (Stage IB) or patient comorbidities. Alternatively, there may be biological plausibility from the interaction of chemotherapy priming the immune response to ICI by causing cell death and increasing antigen exposure. Adjuvant pembrolizumab is currently approved for NSCLC patients regardless of PD-L1 expression, with the prerequisite of at least one cycle of adjuvant chemotherapy.

Both IMpower010 and PEARLS/Keynote 091 show evidence of incremental benefit in DFS with sequential chemotherapy followed by ICI. Patients with stage IIIA disease made up approximately 50% of the population in Impower010 and 30% of the enrolled population in PEARLS/Keynote091. Currently, subgroup analysis does not suggest a differential effect based on stage; however, further follow-up will be necessary to evaluate whether this DFS benefit translates into an OS benefit.

**Table 1.** Trials of adjuvant immunotherapy or targeted therapy in resected NSCLC.

| | Patient | Treatment | Control | Time on Tx | DFS | OS | Approval Indications |
|---|---|---|---|---|---|---|---|
| IMpower010 [22] | IB (>4 cm)-IIIA (7th ed.) Any PD-L1 status | Adjuvant chemotherapy (mandatory) Adjuvant atezolizumab × 1 year | Adjuvant chemotherapy | 1 year | 3 y 60% vs. 48% HR 0.66, 95% CI 0.50–0.88 | | Stage II-IIIA PD-L1+ |
| PEARLS/KEYNOTE 091 [23] | IB (>4 cm)-IIIA (7th ed.) Any PD-L1 status | Pembrolizumab × 18 cycles +/− chemotherapy | Placebo | 1 year | 53.6 m vs. 42 m HR 0.76 (0.63–0.91) | NR | Receipt of at least 1 cycle of adjuvant chemotherapy |
| ADAURA [24] | IB-IIIA EGFR (exon 19 del, exon 21 L858R) | Osimertinib +/− chemotherapy | Placebo | 3 years | HR 0.23 | HR 0.49 (0.33–0.73) 85% vs. 73% | Stage IB (tumours > 3 cm) Within 10 weeks of surgery (no chemo) or 26 weeks if adjuvant chemotherapy. |

Additionally, ongoing studies in the adjuvant setting, such as MERMAID-1, aim to assess the benefit and tolerability of concurrent chemoimmunotherapy to explore the hypothesis of synergy of chemotherapy and immunotherapy. MERMAID-1 is a phase III trial of adjuvant durvalumab concurrently with adjuvant chemotherapy for four cycles followed by durvalumab monotherapy for 48 weeks.

Adjuvant atezolizumab or pembrolizumab are not recommended in patients with tumours containing *EGFR* mutations or *ALK* translocations based on lack of efficacy in the advanced disease setting [25]. However, targeted therapy options for adjuvant systemic therapy now exist for patients with *EGFR* mutations. Earlier trials demonstrated that adjuvant therapy with gefitinib or erlotinib reduced the risk of recurrence but failed to improve OS [26–29]. ADAURA was a phase III trial of osimertinib daily for three years compared to the placebo, in the adjuvant setting for patients with Stage IB-IIIA resected NSCLC with confirmed common EGFR mutations (exon 19 deletion or exon 21 L858R point mutation). Adjuvant chemotherapy was not mandatory but was received by 60% of patients. A significant improvement in DFS was observed for patients randomized to adjuvant osimertinib [24]. The updated analysis confirmed a significant improvement in OS (HR 0.49, 95% CI 0.33–0.73). In stage III disease, the magnitude of benefit was even larger (HR 0.37, 95% CI 0.20–0.64), corresponding to an improvement in five-year OS from 67 to 85%. Adjuvant chemotherapy followed by osimertinib represents a new standard of care for patients with resected stage III NSCLC and common *EGFR* mutations [7].

In summary, adjuvant chemotherapy followed by ICI represents the new standard of care for resected stage II and III NSCLC (Figure 1). In patients with resected stage IB-III NSCLC and common *EGFR* mutations, adjuvant osimertinib should be offered following chemotherapy (in appropriately selected patients). Ongoing trials will help define the role of adjuvant molecularly targeted therapies in other molecularly defined subgroups of NSCLC.

### 2.2. Neoadjuvant Systemic Therapy

Patients with stage III NSCLC detected incidentally at the time of surgery are generally considered for adjuvant systemic therapies described above. However, patients identified with resectable stage III NSCLC at diagnosis are generally considered for some form of neoadjuvant therapy. Multiple treatment approaches have been evaluated over time, including chemotherapy alone or chemoradiation. Previous small, randomized trials suggested improved OS from neoadjuvant chemotherapy [30,31]. Felip et al. reported no differences in OS for neoadjuvant chemotherapy followed by surgery compared with surgery followed by adjuvant chemotherapy [32]. However, patients planned for neoadjuvant therapy were more likely to receive chemotherapy (97% versus 66.2%). Trials evaluating combinations of neoadjuvant chemotherapy and radiation have reported mixed results. A posthoc analysis of patients suitable for lobectomy in the Intergroup 0139 trial reported improved OS in patients undergoing neoadjuvant chemoradiation followed by surgery [4]. However, a similarly designed trial, ESPATUTE, found no benefit to the addition of surgery following chemoradiation [33]. The Swiss Group for Clinical Cancer Research (SAKK) reported no benefit to the addition of radiation to neoadjuvant chemotherapy [34]. Therefore, strategies of neoadjuvant chemotherapy or neoadjuvant chemoradiation emerged as standards of care.

Recently, multiple trials have focussed on the addition of ICI to neoadjuvant chemotherapy in resectable NSCLC, including stage III disease (Table 2). This strategy may improve tumour antigen presentation to the immune system [35] and has demonstrated improved DFS in melanoma patients compared with adjuvant ICI [36]. These trials have utilized pathologic complete response rates (pCR), a surrogate for improved disease outcomes, as important trial outcomes [37]. Historically, pCR rates from neoadjuvant chemotherapy alone are low [38].

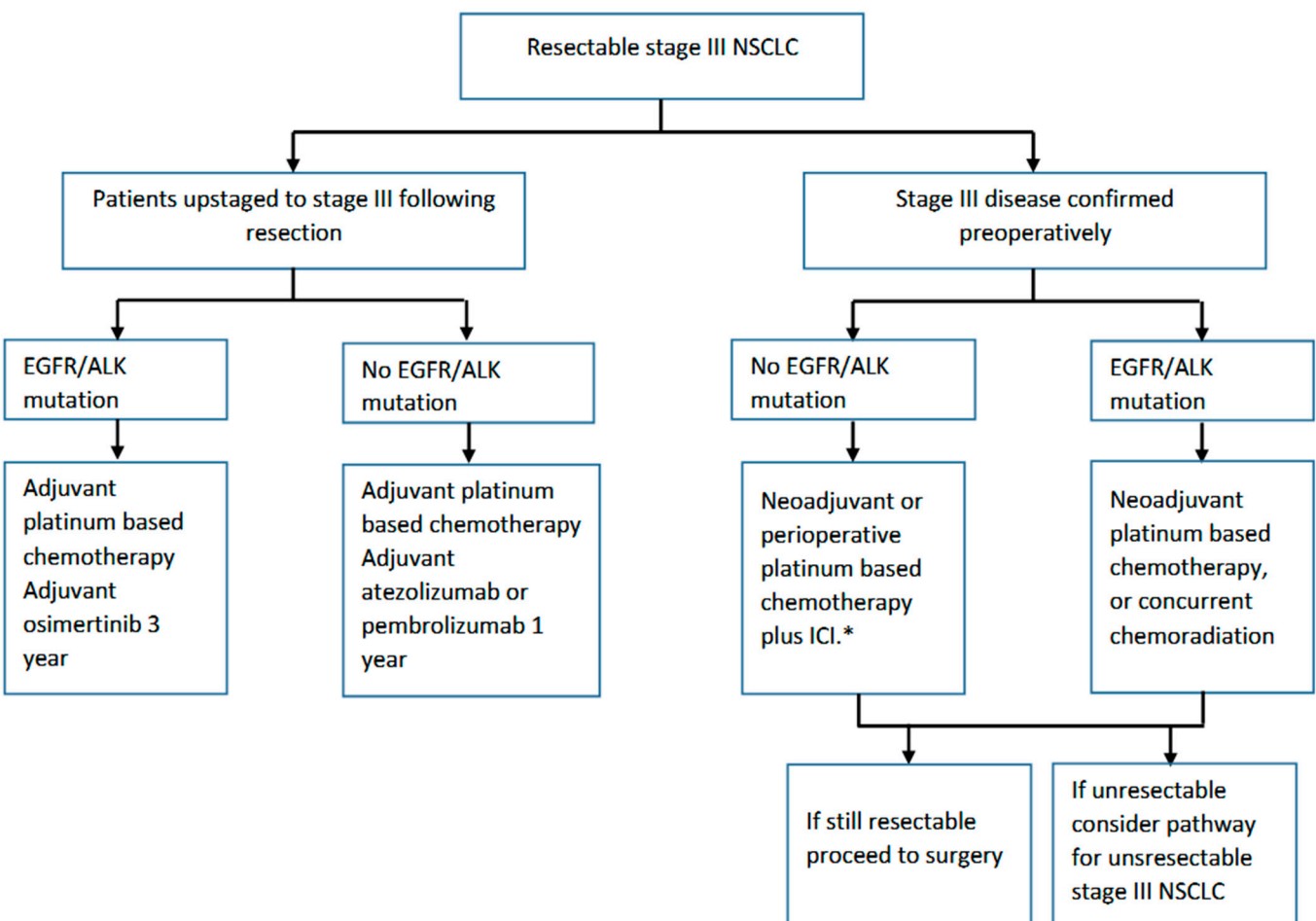

**Figure 1.** Pathway for resectable stage III NSCLC. * ICI—Immune checkpoint inhibitor including nivolumab, pembrolizumab, durvalumab, or toripalimab.

Checkmate 816 evaluated three cycles of neoadjuvant platinum doublet chemotherapy plus nivolumab in resectable stage IB ($\geq$4 cm)—IIIA NSCLC [39]. Surgery was performed within six weeks after the completion of neoadjuvant treatment. There was an option to receive four cycles of adjuvant chemotherapy, and 11.9% in the nivolumab group and 22.2% in the chemotherapy group received adjuvant chemotherapy. The primary endpoint of EFS was improved by approximately 11 months (median 31.6 months versus 20.8 months, HR 0.63, 95% CI 0.43–0.91) with an absolute benefit at two years of 18% (63.8% vs. 45.3%, Table 2). Patients who received chemoimmunotherapy had significantly higher rates of pCR compared with those who received chemotherapy (24% vs. 2.2%). An exploratory analysis suggested EFS was longer in patients who achieved a pCR than in those who did not (HR 0.84, 0.61–1.17) and potentially that the higher rate of pCR drove most of the improvement in DFS observed in the trial. Patients without pCR had similar outcomes as patients receiving neoadjuvant chemotherapy alone; however, the trial was not powered adequately to assess this. Further, follow-up is required to assess if these improvements translate into improved OS.

**Table 2.** Trials of neoadjuvant chemoimmunotherapy in resectable NSCLC.

| Trial | Stage (AJCC 7th ed.) | Treatment | Time on Tx | Comparator | Adjuvant | Time on Tx | EFS | | PCR | | OS | | Surgery | |
|---|---|---|---|---|---|---|---|---|---|---|---|---|---|---|
| | | | | | | | T | C | T | C | T | C | T | C |
| CHECKMATE 816 [39] | IB (≥4 cm)–IIIA | Nivolumab + Platinum doublet × 3 cycles | 9 weeks | Platinum doublet × 3 cycles | Platinum doublet × 4 cycles | No | 31.6 m HR 0.63 (0.43–0.91) | 20.8 m | 24% | 2.2% | HR 0.53 (0.30–1.07) | | 15.6% | 20% |
| AEGEAN [40] | II-IIIB (N2) | Durvalumab + Platinum doublet × 4 cycles | 12 weeks | Platinum doublet × 4 cycles | Durvalumab × 12 cycles | 1 year | NR vs. 63.3% HR 0.68 (0.53–0.88) | 25.9 m 52.4% (2 years) | 17.3% | 4.3% | NR | NR | 20% | 20% |
| KEYNOTE 671 [41] | II-IIIB (N2) | Pembrolizumab + Cisplatin doublet × 4 cycles | 12 weeks | Cisplatin doublet × 4 cycles | Pembrolizumab × 13 cycles | 1 year | NR 62.4% HR 0.58 (0.46–0.72) | 17 m 40.6% (2 years) | 18.1% | 4% | HR 0.73 (0.54–0.99) | | 13.6 | 16% |
| NADIM2 [42] | IIIA-IIIB | Nivolumab + Carbo + pacli × 3 cycles | 9 weeks | Carbo + pacli | Nivolumab 6 months | 6 months | 85% HR 0.47 (0.25–0.88) | 63.6% | 37% | 7% | HR 0.43 (0.19–0.98) | | 93% | 69% |
| NEOTORCH [43] | II-III | Toripalimab + platinum doublet | 9 weeks | Platinum doublet | Toripalimab 13 cycles | I year | 64.7% HR 0.40 (0.277–0.565) | 38.7% | 48.5% | 8.4% | HR 0.62 (0.38–0.999) | | 82% | 73% |

Additional trials have evaluated the combination of neoadjuvant plus adjuvant (perioperative) ICI. AEGEAN evaluated perioperative durvalumab in resectable stage II–IIIB (N2) NSCLC [40]. Patients were randomized to receive four cycles of neoadjuvant durvalumab and platinum-based chemotherapy every three weeks, followed by durvalumab every four weeks for 12 cycles, versus four cycles of neoadjuvant platinum doublet chemotherapy. Results from an interim analysis demonstrated a relative risk reduction of 32% (HR 0.68, 0.53–0.88) in DFS and an absolute improvement of 11% at two years. pCR was similarly improved at 17.3% versus 4.3%, and benefit in EFS was seen across all subgroups, including all PD-L1 expression.

KEYNOTE 671 evaluated perioperative chemoimmunotherapy with pembrolizumab in resectable NSCLC (II-IIIB, N2) with patients randomized to neoadjuvant pembrolizumab and cisplatin-based doublet followed by adjuvant pembrolizumab up to 13 cycles [41]. The study findings were consistent with Checkmate 816 and AEGEAN. There was a significant improvement in DFS (HR 0.58, 0.46–0.72) with an absolute improvement of 12% at two years. A significantly higher pCR rate was observed (18.1% vs. 4.0%). In an exploratory analysis, improved EFS was observed both in patients who achieved a pCR (0.33, 95% CI 0.09–1.22) and those who did not (HR 0.69, 95% CI 0.55–0.85).

Two additional trials, NADIM2 [42] and NEOTORCH [43], also evaluated perioperative nivolumab or toripalimab plus chemotherapy. The findings of both trials were consistent with those above, with improvements in pCR and DFS (Table 2). There are other trials underway with interim analysis results expected within the next 12 months, including Checkmate 77T evaluating perioperative nivolumab and chemotherapy, [44] as well as IMpower030 evaluating neoadjuvant atezolizumab and chemotherapy [45]. Patients with *EGFR* mutations remain excluded from these trials of ICI therapy. However, the, NeoADAURA trial is evaluating neoadjuvant osimertinib with or without neoadjuvant chemotherapy in this setting [46].

The multiple trials of neoadjuvant, or perioperative chemotherapy plus ICI, represent practice-changing data for resectable NSCLC. However, long-term follow-up is needed to determine if improvements in DFS translate into OS benefits. These studies included a high proportion of patients with Stage III disease (60% CHECKMATE 816, 70% KEYNOTE 671). Neoadjuvant or perioperative chemotherapy plus an ICI would now be the preferred approach for resectable stage III NSCLC for the majority of patients unless they have tumours with molecular abnormalities, such as *EGFR* or *ALK*, or have contraindications to the use of an ICI. Similar to the adjuvant setting, there is conflicting data between trials about the importance of PD-L1 expression as a predictive biomarker, and this should probably not be used for patient selection.

It is important to recognize that not all patients with stage III NSCLC benefit from neoadjuvant approaches. There is a risk that patients may not proceed to definitive surgery. In the Checkmate 816, AEGEAN, and Keynote 671 trials, approximately 13–20% of patients in the ICI treatment arm did not undergo surgery. The most common reasons cited included progressive disease and adverse events. However, more patients in the chemotherapy alone arms did not proceed to surgery. This highlights the need for careful consideration with patient selection and upfront assessment of resectability, as well as the involvement of a multidisciplinary panel including thoracic surgeons, pathologists, medical oncologists, and radiation oncologists. Competing treatment strategies for unresectable stage III NSCLC described below should be given consideration in borderline cases. Upfront surgery may be considered for patients who may not be candidates for adjuvant chemotherapy.

Concerns exist that neoadjuvant chemotherapy plus immunotherapy might result in a delay in surgery, intraoperative difficulties (higher incidence of fibrosis, higher conversion rates of VATS to thoracotomy), perioperative morbidity, and mortality. However, this was not observed in the Checkmate 816 trial [39]. More patients in the chemotherapy plus immunotherapy arm had definitive surgery (83.2% vs. 75.4%), and a similar proportion of patients in both arms had delays in surgery. The median duration of surgery was shorter, minimally invasive approaches were used more commonly, and there were fewer pneumonectomies in the chemotherapy plus immunotherapy group. In the Keynote 671 trial, a similar proportion of patients underwent a lobectomy (78.8% vs. 75.1%) [41]. Mortality at 30 days (1.8% vs. 0.6%) and 90 days (2.2% vs. 0.9%) was slightly higher in the chemotherapy plus pembrolizumab group.

It is difficult to directly reconcile benefits between neoadjuvant versus adjuvant strategies of ICI as the trial patients are heterogeneous (predominantly stage IIIA in neoadjuvant studies and stage II in adjuvant), and there are no trials directly comparing these approaches. Neoadjuvant approaches may result in a higher proportion of patients receiving systemic therapy and may be preferred for that reason.

There are many unanswered questions concerning the use of neoadjuvant or perioperative ICI plus chemotherapy. No trials have evaluated the contribution of adjuvant ICI in the setting of neoadjuvant ICI therapy. It is unclear if patients achieving a pCR need an additional year of adjuvant therapy or whether patients with residual disease need consideration of alternate adjuvant systemic therapy, as seen in triple-negative and Her2-positive breast cancers [47,48]. However, markers such as pCR, or minimal residual disease as measured with circulating tumour DNA (ctDNA), may provide an opportunity to refine estimates of prognosis and modify treatment recommendations for patients. An exploratory analysis of clearance of ctDNA from the start of neoadjuvant treatment to the end of treatment in Checkmate 816 found an association between ctDNA clearance and higher rates of pCR and longer EFS [39]. This may potentially help guide de-escalation or intensification strategies in the adjuvant setting.

## 3. Unresectable NSCLC

Unresectable stage III NSCLC is generally characterized by the presence of bulky mediastinal lymph nodes (>3 cm), involvement of multiple mediastinal nodal stations, or the presence of contralateral mediastinal lymph nodes. Historically, unresectable stage III NSCLC was treated with radiation. The Landmark Trial (CALGB8433) and ECOG 4588 both demonstrated that the addition of sequential cisplatin and vinblastine followed by radiotherapy improved median overall survival from approximately 10 to 14 months [49,50]. Five-year OS remained poor. Subsequent trials established concurrent chemoradiation as the standard of care [51,52]. A meta-analysis of trials of concurrent versus sequential chemoradiation showed an absolute improvement in OS of 4.5% at five years [53].

### 3.1. Historical Approach

A variety of platinum-based combination chemotherapy regimens have been utilized in clinical trials of concurrent radiation. The most common of these are cisplatin and etoposide, developed by the Southwest Oncology Group (SWOG) [54], or carboplatin and paclitaxel developed by the Radiation Therapy Oncology Group (RTOG, now NRG Oncology) [55]. While these two regimens have not been compared directly in clinical trials, retrospective analyses suggest similar outcomes [56]. More recently, cisplatin and pemetrexed emerged as an additional option for concurrent chemoradiation in patients with nonsquamous NSCLC. The PROCLAIM trial showed no difference in OS between concurrent cisplatin and pemetrexed versus cisplatin and etoposide in stage III nonsquamous NSCLC [57]. Previous trials of consolidation systemic therapy, either chemotherapy [58] or targeted therapy [59], have failed to improve OS.

Current radiation treatment schedules were established four decades ago. The RTOG 7301 trial established the current radiation dose for patients with stage III NSCLC of 60–63 Gy in 1·8–2·0 Gy fraction sizes [60]. Hyperfractionated and accelerated radiation treatment schedules have demonstrated improved OS in comparison to conventional radiation, but these approaches have not been widely adopted into practice [61,62]. Promising results were observed in early-phase clinical trials in locally advanced NSCLC of dose escalation of radiation therapy up to 74 Gy while reducing irradiated volumes through the use of image guidance and either three-dimensional conformal or intensity-modulated radiation therapy for locally advanced NSCLC [63]. However, the RTOG 0617 trial, a two-by-two factorial phase III trial, comparing dose-escalated radiation to 74 Gy versus a conventional dose of 60 Gy, demonstrated worse OS in the dose-escalated arm (HR 1.38, 95% CI 1.09–1.76) [64]. A second comparison of cetuximab versus no cetuximab showed no improvement in OS (HR 1.07, 95% CI 0.84–1.35). Sixty Gy of radiation remains the current standard of care given concurrently with chemotherapy in stage III unresectable NSCLC.

### 3.2. Recent Advances

The major recent advance in the management of unresectable stage III NSCLC has come from the evaluation of ICI (Figure 2). The PACIFIC trial evaluated the addition of one year of consolidation durvalumab compared to the placebo following concurrent chemoradiation for unresectable stage III NSCLC [65,66]. Durvalumab is a selective, high-affinity, human IgG1 monoclonal antibody that blocks programmed death ligand 1 (PD-L1) binding to programmed death 1 (PD-1) and CD80, allowing T cells to recognize and kill tumour cells. Patients completed radical radiation (54–66 Gy) concurrent with platinum doublet chemotherapy (cisplatin or carboplatin in combination with etoposide, vinblastine, vinorelbine, a taxane [paclitaxel or docetaxel], or pemetrexed). Patients with no evidence of disease progression within six weeks (initially two weeks) of completion of chemoradiation were randomized to receive durvalumab or the placebo. The coprimary endpoints of the study were PFS and OS. The addition of one year of durvalumab consolidation therapy significantly improved both PFS (median 16.8 m vs. 5.6 m, HR 0.52, 95% CI 0.42–0.65) [66] as well as OS [65]. Updated analysis demonstrated improved median OS (47.5 m vs. 29.1 m, HR 0.72, 95% CI 0.59–0.89), as well five-year OS (42.9% vs. 33.4%) [67]. Immune-mediated adverse events of any grade, regardless of cause, were reported in 24.2% of patients in the durvalumab group and 8.1% of patients in the placebo group. Grade 3 or 4 immune-mediated adverse events were reported in 3.4% and 2.6% of patients, respectively. Durvalumab now represents the standard of care following concurrent chemoradiation. Data from a phase II single-arm trial demonstrates similar safety data for durvalumab following sequential chemoradiation [68].

A preplanned analysis was undertaken to determine if PD-L1 expression was predictive of either PFS or OS benefit [69]. PD-L1 expression was assessed using the SP263 antibody with cut-off values of <25% and $\geq$ 25% and was available for 63% of the trial population. PFS has improved significantly in PD-L1 < 25% (HR 0.59, 95% CI 0.43–0.82) and PD-L1 $\geq$ 25% (HR 0.41, 95% CI 0.26–0.65) groups. Similarly, OS was improved in both PD-L1 < 25% (HR 0.89, 95% CI 0.63–1.25) and PD-L1 $\geq$ 25% (HR 50, 95% CI 0.30–0.83). In a posthoc exploratory analysis using cut-points of PD-L1 <1% and PD-L1 $\geq$ 1%, PFS remained improved in the PD-L1 negative subgroup (HR 0.73, 95% CI 0.48–1.11); however, OS was not improved (HR 1.14, 95% CI 0.0.71–1.84). This has resulted in differences in approval by the US FDA (approved independent of PD-L1 expression) and the European Medicines Agency (EMA, approved in PD-L1 $\geq$ 1%). The role of durvalumab in molecularly defined subgroups of NSCLC, such as *EGFR* and *ALK,* remained unclear.

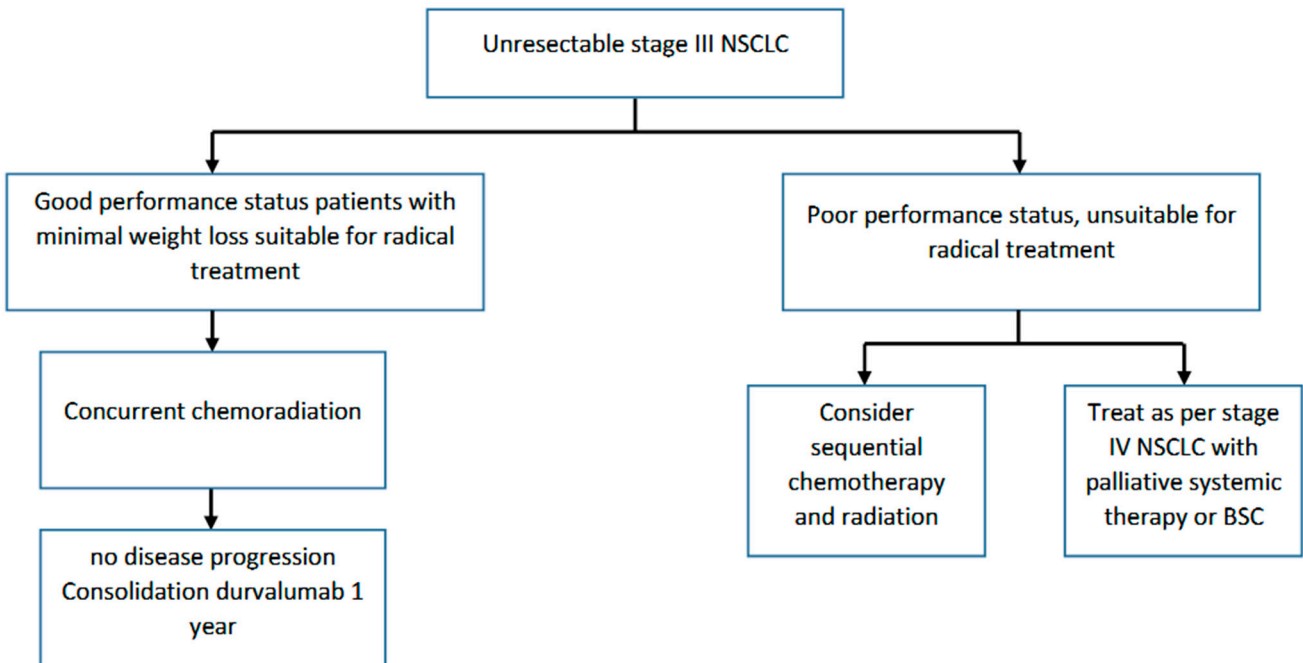

**Figure 2.** Pathway for unresectable stage III NSCLC.

Similar findings were reported from the phase III GEMSTONE-301 trial, evaluating sugemalimab following either sequential or concurrent chemoradiation [70]. PFS was significantly improved (median 9 m vs. 5.8 m, HR 0.64, 95% CI 0.48–0.085). Several single-arm phase II trials have been completed evaluating the addition of immunotherapy to concurrent chemoradiotherapy. These trials evaluated different outcomes, including safety, adverse events, and ORR. Examples include the NICOLAS trial, which evaluated nivolumab [71], the DETERRED trial, which evaluated atezolizumab [72], and KEYNOTE-799, which evaluated pembrolizumab [73]. These trials provide additional safety data, including safety for concurrent administration of ICI with chemoradiation. Additional strategies include the use of dual immunotherapy. BTCRC-LUN16-081 was a phase II trial evaluating dual immunotherapy with ipilimumab and nivolumab versus nivolumab alone for six months following chemoradiation [74]. No difference was observed in median PFS (25.8 m vs. 25.4 m). A randomized phase II trial evaluated the addition of either oleclumab or monalizumab to durvalumab following concurrent chemoradiation [75]. Improvements in PFS were observed for the addition of oleclumab (HR 0.44, 95% CI 0.26–0.75) or monalizumab (HR 0.42, 95% CI 0.24–0.72) to durvalumab. These combinations are being evaluated in the phase III PACIFIC-9 trial [76].

In conclusion, the addition of ICI therapy in stage III unresectable NSCLC has improved treatment outcomes. However, many patients still relapse and die from their disease. This highlights the need for further strategies and novel agents to improve overall survival. Multiple phase II/III trials are ongoing evaluating new agents, and recruitment to these clinical trials is critical to further advancements in our knowledge [77].

**Author Contributions:** A.S., Z.B. and P.M.E. all contributed to defining the question to be addressed in the manuscript, determining the broad outline and writing the manuscript. All authors have read and agreed to the published version of the manuscript.

**Funding:** This research received no external funding.

**Institutional Review Board Statement:** Not applicable.

**Informed Consent Statement:** Not applicable.

**Conflicts of Interest:** A.S. has received honoraria for speaking and advisory board meetings from AstraZeneca; Z.B. has no conflict of interest to declare; P.E. has received honoraria for speaking or advisory board meetings from AstraZeneca, BMS, Eli Lilly, Jannsen, Jazz, Merck, Novartis, Roche, Sanofi, and Pfizer.

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
