# Peer review of "Update on the Management of Stage III NSCLC: Navigating a Complex and Heterogeneous Stage of Disease"

_curroncol, doi:10.3390/curroncol30110689_

Round 1

Reviewer 1 Report

Comments and Suggestions for Authors

The topic of the review is extremely interesting.

The article is well written, very interesting, complete and all the new studies are reported.

Basically, the paper has a logic and clear structure. The title is appropriate for the content of the article. The abstract is concise and summarizes the essential information of the paper. This review is well done, the bibliography complete and the tables well presented. 

Author Response

There are no comments to address from this review

Reviewer 2 Report

Comments and Suggestions for Authors

This is a review of the current /updated treatment recommendations management of stage III NSCLC. Overall the paper is well organized and well written. I would recommend removing any of the extraneous information that is not specifically related to the management guidelines in the 3 different sections. I would also recommend including a flow diagram that summarizes the treatment/management modalities suggested in the paper to tie everything together. The abstract could benefit from some reorganization and more summary info; I would remove references to specific trials since this is an overall review rather than a review of specific trials. 

Comments on the Quality of English Language

No issues.

Author Response

Thank you for the feedback and helpful comments. 

It was a little unclear what the reviewer considered extraneous information. We believe it is important to understand the historical approach to treatment in order to contextualize recent advances. There will be readers from a variety of disciplines and it is important to cover salient information for all these readers.

We have included flow diagrams as requested by the reviewer. 

The review suggested removing references to specific trials as this is an overall review rather than a review of individual trials. We would respectfully disagree with the reviewer on this point. This is an evidenced based review and we believe it is important to include relevant data from the key trials that inform change in practice. Additionally, many readers may not be as familiar with these new trial data and it is important that they understand the design of the trials and the magnitude of benefit, plus any uncertainty about the benefit in key subgroups. We have therefore kept references to specific trials. 

We have included headings in the abstract to address the comment. The abstract is already 259 words and we do not believe there is space to provide additional summary information if we are to stay within the journal guidelines. 

Reviewer 3 Report

Comments and Suggestions for Authors

I would like to congratulate the authors on an interesting manuscript. In their article, the authors discuss and analyze in great detail the results of current research on the treatment of stage III non-small cell lung cancer. The article is very well written, with very good quality English, the authors discuss all relevant literature.

As a thoracic surgeon, I only have a few minor comments on the article:

1. Line 44. The authors state: “the definition of resectable stage III disease is not clearly defined and very surgeon dependent.” As a surgeon, I agree with the statement. However, several studies demonstrated, that decisions made within a multidisciplinary tumor board are associated with better outcomes of lung cancer treatment. I suggest to consider including a brief information on the role of MDT in the decision making process regarding treatment of stage III NSCLC.

2. Surgical operation after ICI treatment may be associated with delay to surgery, intraoperative  difficulties (higher incidence of fibrosis, higher conversion rates of VATS to thoracotomy), perioperative morbidity and mortality. If the authors consider it appropriate, they may briefly discuss these issues.

Once again, I would like to congratulate the authors on a very interesting article.

Author Response

Thank you for the very positive feedback

There is discussion already about multidisciplainry case conferencing and patient selection in the section of neoadjuvant chemoIO. The following sentence has been added in the introduction regarding multidisciplinary case conferencing

In this setting, multidisciplinary case conferencing may improve outcomes.5 It is important to ensure all treatment options are considered and there is agreement from all disciplines, regarding which patients with stage III NSCLC are considered surgically resectable. 

In response to the comment about intraoperative surgical difficulties, we have added the following

Concerns exist that neoadjuvant chemotherapy plus immunotherapy might result in delay to surgery, intraoperative difficulties (higher incidence of fibrosis, higher conversion rates of VATS to thoracotomy), perioperative morbidity and mortality. However, this was not observed in the Checkmate 816 trial. More patients in the chemotherapy plus immunotherapy arm had definitive surgery (83.2% vs 75.4%) and a similar proportion of patients in both arms had delays in surgery. The median duration of surgery was shorter, minimally invasive approaches were used more commonly and there were fewer pneumonectomies in the chemotherapy plus immunotherapy group. In the Keynote 671 trial a similar proportion of patients underwent lobectomy (78.8% vs 75.1%). Mortality at 30 days (1.8% vs 0.6%) and 90 days (2.2% vs 0.9%) was slightly higher in the chemotherapy plus pembrolizumab group.

Round 2

Reviewer 3 Report

Comments and Suggestions for Authors

Thank you very much for considering my suggestions. Once again, I congratulate the authors of the interesting article and I suggest its publication.